# Horse Injury during Non-Commercial Transport: Findings from Researcher-Assisted Intercept Surveys at Southeastern Australian Equestrian Events

**DOI:** 10.3390/ani6110065

**Published:** 2016-10-25

**Authors:** Christopher B. Riley, Belinda R. Noble, Janis Bridges, Susan J. Hazel, Kirrilly Thompson

**Affiliations:** 1Institute of Veterinary, Animal and Biomedical Sciences, Massey University, Tennent Drive, Palmerston North 4442, New Zealand; j.p.bridges@massey.ac.nz; 2School of Animal and Veterinary Sciences, University of Adelaide, Roseworthy SA 5371, Australia; belinda.noble@adelaide.edu.au (B.R.N.); susan.hazel@adelaide.edu.au (S.J.H.); 3Appleton Institute, Central Queensland University, Adelaide SA 5034, Australia; kirrilly.thompson@cqu.edu.au

**Keywords:** horse, equine, float, trailer, truck, injury, driver, transportation

## Abstract

**Simple Summary:**

Research on the transportation of horses has largely focused on the movement of horses by commercial livestock carriers. Information on factors associated with horse injuries sustained during private (non-commercial) transportation in small horse trucks and trailers is limited. This study surveyed drivers transporting their horses to equestrian events in southeastern Australia. Information on drivers, travel practice, vehicle characteristics and horse injury was collected. A total of 55/223 (24.7%) participants reported transportation related injuries to their horses. Of these 72% were described as horse associated (scrambling, slipping and horse-horse interaction), 11% due to mechanical failure, and 6% due to driver error. The risk of horse injury was not associated with driver age or gender, or experience. Drivers that answer the telephone whilst transporting horses, were more likely to report a previous injury experience. There was a trend for participants with <8 hours sleep prior to the survey to have previously had a horse injured. There was a modest positive association between increasing trailer age and the number of injuries. The range of trailer models prevented identification of the importance of individual design features. The study highlights the potential for horses to sustain transportation injuries in privately owned vehicles and warrants further study to address this risk to their welfare.

**Abstract:**

Equine transportation research has largely focused on the commercial land movement of horses. Data on the incidence and factors associated with horse injuries during non-commercial transportation (privately owned horse trucks and trailers) is scant. This study surveyed 223 drivers transporting horses to 12 equestrian events in southeastern Australia. Data collected encompassed driver demographics, travel practice, vehicle characteristics, and incidents involving horse injury. Approximately 25% (55/223) of participants reported that their horses were injured during transportation. Of these 72% were owner classified as horse associated (scrambling, slipping and horse-horse interaction), 11% due to mechanical failure, and 6% due to driver error. Horse injury was not significantly associated with driver age, gender, or experience. Participants that answer the telephone whilst driving were more likely to have previously had a horse injured (*p* = 0.04). There was a trend for participants with <8 hours sleep prior to the survey to have experienced a previous transportation-related injury (*p* = 0.056). Increased trailer age was associated with a greater number of injury reports (r² = 0.20; *p* < 0.04). The diversity in trailer models prevented identification of the importance of individual design features. This study highlights the potential for horses to sustain transportation injuries in privately owned vehicles and warrants further study to address this risk to their welfare.

## 1. Introduction

The use of horses for work, pleasure, sport or commerce necessitates their transportation, often on a regular and frequent basis [1]. Commercial transportation commonly involves the overland movement of mixed groups of animals in large to medium sized single or multi-level trucks over long distances [2,3,4], but may also include movement by sea or air [5]. Previous research on the land transport of horses has focused primarily on welfare considerations for the movement of animals to slaughter or sale (commercial transport) in Europe and the North America [2,3,6,7]. These works have made it clear that multiple stressors may act on horses during transportation, with a cumulative effect on their welfare [4,5]. These include psychological (e.g., confinement, inhibition of eating/drinking) and physical stressors (e.g., temperature alteration, air quality, dehydration) [5,8,9,10,11]. As a result of previous research, it is widely accepted that the transportation of horses and other domestic animals frequently results in adverse welfare outcomes [3,10]. This includes injuries sustained by horses during transportation that result in temporary or permanent loss of function for the animals involved, and a range of effects (both emotional and financial) on those transporting them [5,10,12]. 

In contrast to the commercial movement of horses, owners and drivers in the non-commercial sector generally and frequently transport horses in individual compartments in small two to three horse capacity trailers (called floats in Australia and New Zealand) or trucks (called horse boxes in the United Kingdom), and often over shorter distances for recreational and competitive purposes [5,10,13]. In Australia, many private equine transportation vehicles have a gross vehicle mass <4.5 tonnes, and are therefore not subject to the more stringent regulations that govern the operation of the heavier commercially operated vehicles that are frequently used for livestock movement [14]. Non-commercial horse trailers do not require specific certification of their suitability for use for transporting animals. Whereas there have been many studies of transport-related injuries sustained during the commercial transportation of horses [2,4,12,15,16], there are few peer reviewed publication on the risks of injury during non-commercial transport by owners [12,17]. Equine injuries resulting from commercial transportation have been attributed to horse-associated, vehicle design and driver-associated factors [5,12,18]. However little is known how these factors apply to owners and drivers involved in the non-commercial land transport of horses [8]. The aims of the current pilot study were to document transportation practices and owner reported injury to horses during their non-commercial land movement in southeastern Australia, and to explore the factors associated with any reported injuries.

## 2. Materials and Methods

In accordance with the guidelines provided by the University of Adelaide Human Ethics Committee, a low risk human ethics assessment was completed prior to conducting the survey.

### 2.1. Survey Design

Data on non-commercial horse transportation practices of owner-drivers and related horse injuries were collected via survey tools developed for in-person delivery to volunteer participants on location at recreational and competitive horse sport events. There were separate questionnaires for drivers transporting horses in trucks and those using a tow vehicle with a trailer. Each survey comprised questions that addressed demographic information of drivers, routine towing or truck driving practices, and towing vehicle and horse trailer (or truck) specifications. Those participants who reported having experience of a horse being injured in association with transportation were asked to fill out a separate questionnaire on the details of their horse injury event. Questions soliciting demographic information (*n* = 7) sought responses describing age, gender and driver experience and training. Driving practices whilst towing (*n* = 13) were queried and included the frequency of towing, telephone use, passenger carriage, and the amount of sleep the participant reported in the previous 48 hours. Towing vehicle and trailer (or truck) specifications surveyed (*n* = 16) included information on maintenance schedules, and individual design features. General horse transportation practice data collected (*n* = 12) included the frequency of safety checks and rest periods, the use of protective apparel for horses, the number of animals usually conveyed, breeds and the bodyweights of the horses carried. Finally, data from questions regarding injury events was collected (*n* = 20), including the identification of possible causal factors by respondents, when the injury had occurred, speed of the vehicle at the time of injury, the stage of the journey when the injury occurred (loading, unloading; when moving or stationary), injury location and severity. There were no constraints on participants as to how long ago the incident had occurred relative to the time the survey was conducted. All questions were structured to provide non-open ended responses except in the case where “other” was a response option. 

The performance of the survey tool was piloted at two equine events. The main alteration made after piloting the survey was the exclusion of data collection requiring the measurement of the horse trailer or truck dimensions, as many of the larger equestrian events did not provide access to the vehicles, making collection of these data impractical. Copies of the finalized survey tools that were used are provided in Appendix A.

### 2.2. Survey Data Collection

Data was collected from events in the two most southeastern states in Australia. Surveys were completed by convenience sampling of equestrian events in South Australia and Victoria from September 2012 to March 2013 (Table 1). Events ranged from junior, novice and amateur level through to senior, experienced and professional levels of participation. The majority of surveys were collected in the state of South Australia within 100 km of the city of Gawler, encompassing the capital city (Adelaide). Some surveys were collected at a national event in Melbourne, Victoria. A sampling frame of 300 surveys was determined based on the estimated number of interviews that could be undertaken per event, and the calendar of scheduled equestrian events in the region accessible for sampling. Drivers of horse trailers or light trucks that had transported horses to the events were invited to voluntarily participate in the survey, with an aim to capture approximately 15% of drivers per event. Surveys were administered by a face-to-face interview. Four people were trained to administer the survey to reduce individual interviewer bias. The surveys required approximately 15 to 20 minutes per participant to complete.

### 2.3. Statistical Analysis

Data were analyzed using R software (R v. 3.0.2, Foundation for Statistical Computing, Vienna, Austria). Chi squared tests were calculated to determine the significance of associations (if any) between study variables and injury. The effect of each dichotomous variable on the outcome (injury or non-injury) of interest was determined by the Mann-Whitney U test. Categorical variables were evaluated by Kruskall-Wallis non-parametric ANOVA. The effect of normally distributed parametric variables on injury risk was evaluated with one-way ANOVA. Some categorical variables (e.g., average distance travelled) were transformed into two or three classes and Chi squared tests were calculated to determine associations between them and injury (refer Appendix A for further information on survey variables). For the survey question on the trailer tow vehicle: “Which of the following is included in your routine check”, the dichotomous responses to each subcategory in the question were summed (each item being assigned a value of one if the check items had been performed; zero if not) to create a “Vehicle safety check score”, and analyzed as described above. The level of statistical significance for all statistical tests was set at *p* < 0.05 with a trend indicated at *p* < 0.1. Multivariate analysis was explored for this data set, with variables considered for inclusion in a binary logistic regression model to predict the occurrence of horse injury (yes/no). Variables that were identified as potential factors and suitable for entry into the model were based on the results of Chi square tests, where *p* < 0.2. All predictors were assessed for collinearity prior to final model development.

## 3. Results

### 3.1. Descriptive Data

There were 223 participants surveyed at 12 equestrian events between September 2012 and January 2013. Descriptive statistics for non-normally distributed (median, interquartile range, and range) variables are given in Table 2. There were 157 female participants, 62 males, and four of undisclosed gender. The frequency of horse transportation reported by drivers and the median of the average distance travelled per trip is given in Table 2. There were 46 different manufactured brands of trailers (168/193 trailers), nine homemade trailers, and 16 with an unreported origin. Characteristics of the trailers used by participants are listed in Table 3. Straight load trailers (two horse, side by side) were the most commonly represented type, followed by angle load and gooseneck trailers. Of the 30 trucks, there were 12 models: Hino (8/30), Isuzu (7/30), Ford (3/30), Iveco (3/30), Mitsubishi (2/30) and one each of seven other manufacturers. The trucks generally had two (20/30) or three axles (9/30), and ranged in age from new to 42 years.

### 3.2. Horse Injuries

Of the participants, 55/223 (24.7%) reported incidents in which their horses had been injured during transportation in a trailer (22.8%; 44/193) or small truck (36.7%; 11/30). Over half of the injuries involved the lower limbs (hind limbs 33.9%; forelimbs 22.6%), with the head and muzzle (14.5%), chest (9.7%), flank/hindquarter (9.7%), neck (6.5%) and tail (3.2%) less frequently reported sites of trauma. The median time that had elapsed between the time of the survey and the recalled trailer incident was available for only 35/44 participants. Of these 15/35 (43%) had occurred within three years prior to the survey, 7/35 (20%) within 3 to 10 years, 4/35 (11%) within 10 to 15 years, and 9/35 (26%) between 15 and 20 years. Participants attributed injuries to single or multiple factors.

### 3.3. Owner-Reported Factors Associated with the Risk of Horse Injury

#### 3.3.1. Transportation Vehicle Associated Factors

There was no significant association found between the frequencies of horse injuries and transportation by a trailer or small truck, but there was a trend for a higher injury rate associated with trucking (*p* = 0.10). Most injuries were reported to have occurred while the loaded transportation vehicle was moving (83.6%; 46/55), with fewer occurring during unloading (7.3%; 4/55), loading (3.6%; 2/55) and while the loaded vehicle was stationary (3.6%; 2/55). Trailer-associated injuries usually occurred during vehicle movement (86.4%; 38/44), and less frequently during unloading (6.8%; 3/44), loading (2.3%; 1/44) and while stationary (4.5%; 2/44). Truck-associated injuries usually occurred during vehicle movement (72.7%; 8/11), and less frequently during unloading (9.1%; 1/11), and loading (9.1%; 1/11). No injuries were reported in trucks while stationary. For one truck related injury, the timing with respect to the journey was not described. Mechanical failure of a trailer component accounted for a tenth of injuries, and included descriptions of rotten flooring, the hitch point snapping, and failure of the suspension (Table 4). 

As the subsample size for each trailer brand was limited, the rate of injury with respect to specific trailer brands was compared only for participants with the three most common models: metal Horseman^®^, and Olympic^®^ trailers, and wooden Taylor^®^ trailers; differences among these brands were not statistically significant. The horse carrying capacity of the vehicle (the number of horses that owners reported could be loaded into the vehicle) was associated with a trend towards an increase in the risk of injury (*p* = 0.09); i.e., those who reported a higher number of horses could be transported in their trailer were more likely to report a horse injury. Information on the age of the trailer was available for 177 of 193 trailers. Following removal of trailer ages for which no injuries were reported, there was a modest linear relationship between trailer age and the percentage of horse injuries reported by respondents (r² = 0.20; *p* = 0.04). 

#### 3.3.2. Driver Associated Factors

Traffic related causes were described by drivers to account for less than a tenth of injuries, and occurred most commonly as a result of sudden braking or turning to avoid other vehicles that had stopped or turned suddenly (Table 4). An admission of driver error accounted for approximately 6% of horse injuries. The use of a vehicle with an insufficient tow rating, overloading a trailer and failure to untie a horse’s head when unloading were reported as examples (Table 4). A small percentage of injury events were attributed to road conditions. Participants who on average travelled greater than 200 km to an event were more likely to report previously having a horse sustain an injury during transportation (*p* = 0.04). Answering the telephone while driving was associated with the participant having previously experienced an incident involving injury to a horse during trailer or truck transportation (*p* = 0.04). There was a trend for participants who had slept less than 8 hours in the night before the survey to be more likely to report having previously experienced an incident involving injury to a horse during trailer or truck transportation (*p* = 0.056). Participants reported that the injuries occurred either between 6 a.m. and 9 a.m. (15/55 incidents), 9.01 a.m. and 12 noon (16/55), 12.01 p.m. and 3 p.m. (12/55), 3.01 p.m. and 6 p.m. (10/55), or after 6 p.m. (1/55); one respondent did not indicate a time of day. The mixture of descriptive (e.g., morning) and quantitative reports for time of day prevented analysis to evaluate a temporal effect on the occurrence of injury. Many drivers reported that they did not conduct a thorough mechanical inspection prior to each journey. There was a trend only (*p* = 0.08) for lower vehicle safety check scores to be associated with participants who reported an accident. Participants who did not check hydraulic brake fluid on trailers also showed a trend (*p* = 0.07) for reporting an injury; i.e., those who did not check were more likely to report an equine injury.

#### 3.3.3. Horse Associated Factors

As described by drivers, horse-associated factors accounted for most injuries (Table 4). Behaviors such as “scrambling” and “panicking” were the most frequently described in this category; “scrambling” was reported to occur most commonly during cornering (7/16). The risk of injury among the three most commonly represented breeds (Arabian, Thoroughbred and Warmblood) and their derivatives were compared; differences were not statistically significant (*p* > 0.05).

#### 3.3.4. Binary Logistic Regression

The results of the questions relating to current driver behaviors and practice, and whether not they had reported a previous experience with horse injury during transportation (No/Yes) was explored in a binary logistic regression. The output from a logistic regression model describing the relationship between these factors and report of a previous horse injury is shown in Table 5. The explanatory variables ultimately selected were likelihood of answering phone and average distance travelled while transporting horses (<100 km, 100–200 km and >200 km). There were an insufficient number of survey participants (power) to account for other variables in the model.

## 4. Discussion

This pilot study found one in four participants (~25%) had experienced an injury to a horse during transportation, confirming that the non-commercial vehicular movement of horses poses a welfare hazard similar to that reported for commercial transportation to slaughter [2,3,4]. 

The median number of journeys and the distances described in the survey are arguably determined by the driver’s equestrian activities and preferences [1]. These are difficult to separate for the purposes of factor analysis, creating a “risk event” for horse injury that brings together interactions among driver, horse and vehicle factors [10]. A recent report on the transport of inexperienced horses on journeys of short temporal duration to slaughter found as many as 44% of adult horses were injured, largely attributable to poor management practices [16]. In contrast, the median number of journeys in the current study was greater, presumably increasing the cumulative risk exposure, particularly for unloading and loading injuries. This can be offset by habituation to transportation that is regarded as a moderator of horse injury risk [10,19]. Furthermore there is evidence of superior welfare outcomes for horses during loading when positive, rather than negative reinforcement training (habituation) techniques are used [20]. However, in an example of “risk homeostasis”, some drivers may be less vigilant when transporting experienced horses or ones that they consider to be “good travelers” [21]. Further objective investigation of the effects of training on the risk of horse injury during trailer transportation is warranted. Future self-report surveys should ask about the type of training approach used by the person typically loading the horse. The median journey distance was generally less than that of recent reports for commercial movement, but the higher risk of injury during longer journeys (>200 km) is in agreement with findings of these commercial studies despite marked differences in livestock transportation practices [4,15,22]. 

The anatomic distribution of injuries provides some evidence to underpin the investigation of measures for injury prevention. The predominance of limb injuries found in this study is in agreement with a recent report of 13 horses injured in road traffic accidents (RTA) during transportation [23]. However a greater percentage of horses had concurrent fore and hind limb trauma, head and neck injuries in the RTA study [23]. Findings in the latter study reflect the focus of the study on hospital-based records of traffic related injuries, frequently initiated at high speeds [23]. Several authors (and some manufacturers) have touted the value of protective equipment to prevent equine injury during transportation, but there is a lack of objective evidence supporting their use [10,24]. The current study sought to determine the possible protective value of this equipment, but there was insufficient statistical power to demonstrate such effects. A protective effect of “shipping boots” on the limbs was not proven statistically, but the high proportion of limb injuries and their consequences supports their use as a reasonable measure to moderate the severity of injuries [24]. 

The range of manufactured brands of trailers and trucks was diverse, and significant injury rate differences among them not statically confirmed. The breadth of types is not a surprising finding given that there are currently no regulations that specifically govern the design and manufacture of trailers or trucks for the non-commercial haulage of horses in Australia. In contrast to the situation for commercial vehicles, the regulation of non-commercial horse trailer (trailer) design is equally lacking internationally [10,25]. It is unclear to the authors why the design and operation of all vehicles specifically used for animal transportation (commercial and non-commercial) in Australia are not governed within a regulatory framework. Such diversity in design made the evaluation of individual trailer features, and differences in the risk of injury attributable to the brand or manufacturer non-feasible from a statistical perspective. This is a challenge likely to be encountered by other researchers, with few studies addressing the issue of design to date [2,26,27,28]. It is suggested that the manipulation and evaluation of trailer design features may be more feasibly evaluated using computer based modeling of equine transportation, underpinned by objective measurements of the biomechanical and behavioral responses of horses to the dynamic mechanical environment of the moving vehicle. These techniques have been applied to some degree in high capacity livestock trailers [29,30], but the authors are aware of only one conceptually relevant study that has sought to model the mechanical stresses encountered during the transportation of horses [31]. 

The movement of horses to slaughter in non-articulated livestock trucks has been associated with a high rate of injury [16]. In the current study the rates of injury reporting were not significantly higher for those that move horses in trucks than those that use trailers. A survey of a larger sample of non-commercial horse transport is warranted to verify this finding. Mechanical failure of a trailer component accounted for a number of injuries (11%), and there was modest association between increased trailer age and injury. The importance of this finding may be associated with trend towards a lack of vehicle inspection/maintenance and the risk of horse injury, or reflect different design features in older vehicles. It is unclear based on the survey data if this subgroup of related injuries were preventable, but the training of drivers in vehicle safety inspection, and the employment of a safety checklist prior to the loading of horses are recommended as reasonable steps towards mitigating this risk [10]. 

Driver experience, training and behaviors are critical to the welfare of transported animals [5,10,30]. A number of driver-associated factors surveyed were either directly associated with an equine injury event, or pertained to behaviors and practices that pose risks to the horses being moved. Age and experience were not found to mitigate the risk of horse injury. This finding is in contrast to a recent review of professional heavy vehicle drivers, with risk of RTA and fatality involvement declining with increasing age from 27 years of age until 63 years [32]. The current survey was not designed to explore causal effects for individual incidents in a high level of detail, but risk behaviors may be recurrent. Very few drivers surveyed admitted to an error on their part being associated with horse injury. However the patterns of equine injury and driver behaviors (below) point to an underestimation of their role in the risk of transportation related injury. The authors recommend further case study research on individual incidents using critical incident interviewing techniques to better understand the significance of diver-associated factors associated with animal injury during non-commercial transportation.

Admitting the practice of answering the telephone while driving, a source of driver distraction, was associated with participants who reported experiencing a previous transportation related injury to a horse. The use of mobile telephones and devices whilst driving is well recognized as increasing the risk of a RTA in normal driving situations [33,34], as it delays recognition and reactions to unexpected events while driving [35]. This specific form of distraction has been linked to a 4.9-fold increase in the risk of a RTA in Australia [34]. Other driver distractors including carriage of passengers and pets, eating and drinking have been shown to increase the risk of a RTA [33,36,37], but were not significantly associated with horse injury in the present survey. However, it must be noted that none of the horse injury incidents reported in this survey occurred as a result of a RTA. Nevertheless it is likely that driver behaviors resulting from inattention do contribute to horse injury risk as a result of rapid correction maneuvers (e.g., swerving or braking suddenly), impacting the horse’s ability to maintain balance [10].

Fatigue is frequently associated with the risk of a RTA for truck drivers, impairing alertness, situational awareness, reaction times, vigilance and decision-making [35,38,39]. The trend for an association between participants reporting <8 hours sleep the night before the survey and reporting a previous transportation related horse injury does not demonstrate causality. It does however provide evidence that people driving to equestrian events may be subject to fatigue. A study of 415 commercial farm livestock truck accidents in North America found 85% of incidents were associated with driver error, 80% were single vehicle accidents, and fatigue was implicated as the major contributor to subsequent livestock fatalities and injury [39,40]. Based on the evidence of possible sleep deprivation in the current study, the role of fatigue should be evaluated in any future factor analysis of injury to horses during non-commercial transportation. Other possible sources of driver error such as alcohol and drug abuse, illness or emotional distress were not investigated in the current study [34]. 

Traffic-related factors reported by participants accounted for less than 10% of injuries, most commonly as a result of sudden braking. The extent to which driving practice versus road conditions contributed to these injuries is unclear, but driver training and education are likely to decrease the need for sudden braking and accidents that result in horse injury during transportation [10,30,34]. The introduction of legislative requirements in the form of, for example, a towing license such as that implemented in the United Kingdom for towing larger trailers by the public, may be beneficial in developing a baseline of knowledge and skills amongst the towing population. The United Kingdom currently requires the certification of drivers transporting horses for commercial purposes [41]. The associated training and workbook available through the British Racing School and other providers may be of value in educating the currently exempt non-commercial drivers domestically and abroad [42]. 

The injury rate among horse breeds did not differ significantly, a finding supported by a study that failed to find breed differences for behavioral problems during trailering [17]. Scrambling and loss of balance were identified by participants as being associated with more than half of the injuries. Although drivers attributed these to their equine passengers, these responses are usually associated with vehicular turning and braking [17,18,24]. The tendency for participants to attribute transportation-related injury to horse-associated factors (i.e., to an external locus of control) contrasts with event riders that take responsibility for horse injuries or accidents (i.e., an internal locus of control) [43]. Further research is needed to determine the extent to which drivers contribute to scrambling and imbalance. Notwithstanding the need to optimize equine trailer or truck design, the effects of turning and breaking can be reduced by better training of non-commercial drivers to anticipate road and traffic conditions in consideration of their live cargo [24]. 

A number of limitations are evident based on the study design, and the sample size with respect to the number of injuries reported. The direct interview method was chosen in an effort to reduce self-selection bias and information bias [44]. This was to ensure that participants included those who had and had not experienced a transportation related horse injury. However, a predefined risk period (e.g., 2 years prior to the survey) was not set, increasing the effect of recall bias on these data [45]. As the range of time that had elapsed between the survey and a reported horse transportation injuries varied from one to 1 to 12 years, it is likely that this source of bias was significant, and true risk could not be accurately calculated. The high number and degree of variability among categorical variables, and the limited power of the study based upon the number of injured horses (*n* = 44 for trailer-related injuries) were also significant obstacles to meaningful multivariable analyses. Increasing the number of survey participants to describe more injured horses may require alternative survey approaches with a more tightly defined recall period, accepting that each approach comes with its own sources of bias [44].

## 5. Conclusions

The findings of the current study provide background information that may assist in the design and focus of future studies of non-commercial transportation of horses. More refined tools for collecting the data are required to better understand the contributing factors to accident and injury in non-commercial horse land transportation. The number of injuries resulting from the recollections of the participants over time is concerning, but should not be mistaken for an indication of period prevalence. The survey of a larger number of participants is recommended to better determine the risk of injury, and thus the wider welfare impact of non-commercial transportation-related injuries. Attention should also be focused on the role of causative factors that underpin transportation related injury so that measures can be taken to mitigate this risk to equine welfare.

## Figures and Tables

**Table 1 animals-06-00065-t001:** Southeastern Australian equestrian event sampling schedule.

Event	Date	No. of Drivers Sampled		Total No. of Surveys
		Trailers	Trucks	
Adelaide Royal Show	7–10/09/2012	42	18	60
Gawler River Pony Club (Eventing)	23/09/2012	14	1	15
Endurance State Championships, Monarto ^1^	29–30/09/2012	23	2	25
Endurance Event, Mount Crawford ^1^	21/10/2012	2	0	2
Horse of the Year (Show horse), Mount Pleasant ^1^	21/10/2012	11	2	13
Cattle Clinic (Campdrafting), Mount Pleasant ^1^	21/10/2012	5	2	7
Northern Hills Pony Club, (Gymkhana) ^1^	28/10/2012	11	0	11
Kapunda Agricultural Show ^1^	03/11/2012	18	2	20
Arabian State Show ^1^	10/11/2012	18	1	19
Williamstown Pony Club (Dressage) ^1^	11/11/2012	20	1	21
Equitana, Melbourne ^2^	4–17/11/2012	20	1	21
Mount Lofty Pony Club (Eventing) ^1^	03/03/2013	9	0	9
Total		193	30	223

^1^ South Australia, Australia; ^2^ Victoria, Australia.

**Table 2 animals-06-00065-t002:** Descriptive statistics for driver, vehicle, journey, and horse-associated variables.

Survey Variable	*N* ^1^ (%)	Median (IQR)	Range
Driver age (years)	185 (83)	46 (12)	18 to 76
Driving experience (years)	171 (77)	29 (17)	2 to 57
Distance traveled to this event (km)	115 (56)	50 (75)	0 to 700
Average distance per trip to events/activities (km)	187 (84)	100 (100)	10 to 1300
Number of trips per annum	169 (76)	36 (30)	1 to 365
Towing experience (years)	171 (77)	20 (20)	0 to 55
Number of passengers typically carried	192 (86)	2 (1)	0 to 5
Time between rest breaks for the driver (h)	191 (86)	3 (1)	0 to 8
Amount of sleep if ≤8 h in previous 24 h (h) ^2^	185 (83)	7 (2)	0 to >8
Amount of sleep if ≤16 h in previous 48 h (h) ^3^	190 (85)	14 (4)	0 to >16
Vehicle safety check score	221 (99)	4 (2)	0 to 6
Protective equipment use score	220 (99)	2 (1)	0 to 4
Horse vehicle carrying capacity	219 (98)	2 (0)	1 to 14
Number of horses normally transported	219 (98)	2 (1)	1 to 12
Maximum horse height (hands)	221 (99)	16–17 (n/a)	<14 to >17
Horse weight usually transported (kg)	221 (99)	500–1000 (n/a)	<500 to >3500

^1^ Number of responses out of 223 total participants; ^2^ Four additional drivers reported ≥8 hours of sleep; ^3^ Four additional drivers reported ≥16 hours of sleep.

**Table 3 animals-06-00065-t003:** Descriptive statistics for trailer-associated variables **^1^**.

Variable ^1^	*N* ^2^ (%)	Median (IQR)	Range
Trailer age (years)	177 (92)	11 (21)	0 to 51
Trailer type	189 (98)	straight load	
Maximum horse capacity	188 (97)	2 (0)	1 to 5
Braking system type	190 (98)	electric	
Suspension type	173 (90)	leaf spring	
Number of axles	188 (97)	2	1 to 3
Hitch tow weight rating (kg)	177 (92)	1500–3000	<1500 to >6000
Chest bar type	189 (98)	moveable	
Breech closure type	189 (98)	small bars	
Bay divider type	188 (97)	partial	

**^1^** For a full description of feature options please see Appendix A describing the survey tools; **^2^** Number of responses out of 193 total trailer-towing participants.

**Table 4 animals-06-00065-t004:** Driver reported factors associated with horse injury sustained in association with the non-commercial transportation of horses.

Causal Factor	Percentage of Reported Injuries
Horse associated (total)	72.7 (40/55)
● Scrambling ^1^	40 (16/40)
● Panicking ^1^	30 (12/40)
● Loss of balance ^1^	12.5 (5/40)
● Loading/unloading problems ^1^	12.5 (5/40)
● Horse-horse conflict ^1^	5 (2/40)
Mechanical failure of a trailer	11 (6/55)
Traffic related causes	9 (5/55)
Driver error	5.5 (3/55)
Road conditions	1.8 (1/55)

^1^ Values given as a proportion of owner reported horse associated factors.

**Table 5 animals-06-00065-t005:** Contributions to the risk of injury in a binary logistic regression multivariable model.

Variable	Coefficient (β)	Standard Error	*z* Value	*p* Value	Odds Ratio	95% CI
Intercept	−1.95	0.33				
Answer phone while driving	0.71	0.36	1.95	0.052	2.03	0.99–4.16
Distance travelled 100–200 km	0.72	0.41	1.75	0.081	2.05	0.92–4.65
Distance travelled >200 km	1.01	0.47	2.17	0.030	2.76	1.09–6.93

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
