# Peer review of "Horse Injury during Non-Commercial Transport: Findings from Researcher-Assisted Intercept Surveys at Southeastern Australian Equestrian Events"

_animals, 2016, doi:10.3390/ani6110065_

Round 1

Reviewer 1 Report

This paper provided data in an under-explored area of study; injury-risk for horses being transported in non-commercial transport. As data in this area are sparse it is particularly helpful for those interested in this area to have access to such data. The paper was well written and easy to read. It was well referenced, and followed an appropriate format.

The only disappointment I had in the paper was that the sample size (for those whose horses had received injury) was insufficiently large to enable more rigorous statistics to be applied. However, I appreciate the logistical and financial constraints of the project and feel that the overall sample size was sufficient to warrant publication.

I have one major revision that I feel needs to be addressed and a number of minor revisions/typos

Major revision

One issue that I think needs to be tightened up in the manuscript is the accuracy of language when reporting on the variables in the questionnaire e.g. Line 24. “Drivers answering the telephone whilst transporting…”. Referring to the questionnaires used in the study there is a question that asked (generally) what the likelihood was of drivers ‘making’ phone calls, and in the accident-specific questionnaire there was a question about whether the respondent felt that ‘distraction due to phone’ was a factor in the accident. Neither of these clearly or directly indicate that the driver was ‘answering the telephone’, so I feel that currently you are not being sufficiently rigorous in how you are describing driver behaviour. It’s not clear to me which variable you are referring to in Line 24 (and probably elsewhere in the manuscript) – as this is one of the main findings in your study I think you need to reflect more accurately the wording of the question you are referring to. In the methods section Lines 80-90 I think you should make it clearer that there were two variants of the questionnaire (one for floats and one for trucks) and an additional questionnaire for those whose horse had sustained an injury during transport. You should also add that the former questionnaires included questions about the journey on the day and ‘typical’ habits/behaviours (or however you want to refer to the contents) and that the accident questionnaire referred to the specific details surrounding the accident. I think it is important to be specific/accurate here and it is a strength in the study. That said, I think you do need to look over the manuscript and check that you are clear when you are reporting specific factors about the occasion of the accident (from the accident questionnaire) and relating them as risk factors for horse injury, and where you are reporting generalised factors from the truck/float survey about the journey on the day of the survey and relating them to the accident. I think these are different classes of variable and should be more clearly presented so that readers can accurately interpret the findings.

Minor revisions and typos.

Line 16, Simple summary – there is an ‘I’ before the word movement at the end of the line, this needs to be deleted.

Line 27. Could the authors find a different phrase to ‘trailer makes’ when I read this I saw ‘makes’ as a verb in the sentence and it was confusing to read correctly. Could the term ‘models of trailer’ or something similar.

Line 28. Word missing in this sentence after ‘owned’ (transport?) and warrant should be changed to ‘warrants’.

Line 71-73. I would have liked to have been told more about the research findings/results of these commercial transport studies. I think it would be better if some details were included here so readers can make some easy comparisons/appreciate differences in the current study.

Line 79. Is an ethics reference number required?

Line 140 (and elsewhere) I was unsure of the use of the word ‘surveys’ when I would generally refer to questionnaires as the instrument used to collect data and ‘survey’ for the overall methodology. i.e. not refer to ‘223 surveys collected’, rather ‘223 questionnaires’. This may be a difference between US/UK English use/terminology, and a discretionary revision.

Line 150. ‘seven’ not ‘7’ – usually numbers 1-10 are written in full in text.

Table 3. I felt that some of these categorical variables, e.g. trailer type, braking system type and other ‘type’ variables, would be better presented/easier to interpret if shown in their categories/frequencies – rather than medians. Even if it was easier just to show the top three most frequently reported (rather than all types, if there are many.)

Line 217. The figure 7/16 would seem to refer to just one of the horse behaviours (scrambling?) rather than both scrambling and panicking – which are currently the subject of the sentence. This sentence should be revised either to reflect both behaviours (eg x/28) or just refer to scrambling.

Line 239. Are you suggesting that there is less representation from SA and Vic because of a (relative) lack of internet connection? I’m not sure I agree, if this is what you are saying. Perhaps consider reviewing this sentence or make it clearer what point you mean here.

Line 330. Typo ‘to the on the dashboard’

Author Response

Thank you for your assistance in improving this manuscript. Please find a summary of our specific responses below. 

This paper provided data in an under-explored area of study; injury-risk for horses being transported in non-commercial transport. As data in this area are sparse it is particularly helpful for those interested in this area to have access to such data. The paper was well written and easy to read. It was well referenced, and followed an appropriate format.

The only disappointment I had in the paper was that the sample size (for those whose horses had received injury) was insufficiently large to enable more rigorous statistics to be applied. However, I appreciate the logistical and financial constraints of the project and feel that the overall sample size was sufficient to warrant publication.

Response: The authors have made further comments in the discssion to recognise the limitation of the current study.

I have one major revision that I feel needs to be addressed and a number of minor revisions/typos

Major revision

One issue that I think needs to be tightened up in the manuscript is the accuracy of language when reporting on the variables in the questionnaire e.g. Line 24. “Drivers answering the telephone whilst transporting…”. Referring to the questionnaires used in the study there is a question that asked (generally) what the likelihood was of drivers ‘making’ phone calls, and in the accident-specific questionnaire there was a question about whether the respondent felt that ‘distraction due to phone’ was a factor in the accident. Neither of these clearly or directly indicate that the driver was ‘answering the telephone’, so I feel that currently you are not being sufficiently rigorous in how you are describing driver behaviour. It’s not clear to me which variable you are referring to in Line 24 (and probably elsewhere in the manuscript) – as this is one of the main findings in your study I think you need to reflect more accurately the wording of the question you are referring to. In the methods section Lines 80-90 I think you should make it clearer that there were two variants of the questionnaire (one for floats and one for trucks) and an additional questionnaire for those whose horse had sustained an injury during transport. You should also add that the former questionnaires included questions about the journey on the day and ‘typical’ habits/behaviours (or however you want to refer to the contents) and that the accident questionnaire referred to the specific details surrounding the accident. I think it is important to be specific/accurate here and it is a strength in the study. That said, I think you do need to look over the manuscript and check that you are clear when you are reporting specific factors about the occasion of the accident (from the accident questionnaire) and relating them as risk factors for horse injury, and where you are reporting generalised factors from the truck/float survey about the journey on the day of the survey and relating them to the accident. I think these are different classes of variable and should be more clearly presented so that readers can accurately interpret the findings.

Response: The manuscript has been reworded to ensure that the tendency to answer the phone was not directy linked to the report of the injury cases, and its assocaition as a behaviour clarified.  Changes have been made to this effect in the summary, results and discussion sections.

Response: Additional information about the questionairre indicating separate truck, trailer and injury modules has been added in the survey design section of the methods and materials section (in support of the supplementary materials).

Response: Further details suggested by the reviewer regarding the questionairres have been provided by inclusion of the original questionairres in the supplementary materials provided with  the manuscript.

Response: As suggested by the reviewer the authors have  made adjustments in the manuscript to improve the clarity around the issues raised by him/her.

Minor revisions and typos.

Line 16, Simple summary – there is an ‘I’ before the word movement at the end of the line, this needs to be deleted.

Response: Adjusted in text – thank you

Line 27. Could the authors find a different phrase to ‘trailer makes’ when I read this I saw ‘makes’ as a verb in the sentence and it was confusing to read correctly. Could the term ‘models of trailer’ or something similar.

Response: Change made to text as requested

Line 28. Word missing in this sentence after ‘owned’ (transport?) and warrant should be changed to ‘warrants’.

Response: Changes have been made in text

Line 71-73. I would have liked to have been told more about the research findings/results of these commercial transport studies. I think it would be better if some details were included here so readers can make some easy comparisons/appreciate differences in the current study.

Response: The authors considered this request carefullly. The introduction has been carefully constructed to outline the research questions, and  is not intended as a comprehensive review of equine transportation. The focus of paper is on non-commercial transport, and many of the key papers including some recent reviews of equine transportation are cited in the reference list.

Line 79. Is an ethics reference number required?

Response: The Human Ethics Committee at the University of Adelaide does not issue a number for ethics approvals of this type.

Line 140 (and elsewhere) I was unsure of the use of the word ‘surveys’ when I would generally refer to questionnaires as the instrument used to collect data and ‘survey’ for the overall methodology. i.e. not refer to ‘223 surveys collected’, rather ‘223 questionnaires’. This may be a difference between US/UK English use/terminology, and a discretionary revision.

Response: This statement has now been re-expressed in response to the reviewer’s concern

Line 150. ‘seven’ not ‘7’ – usually numbers 1-10 are written in full in text.

Response: Changed to “seven”

Table 3. I felt that some of these categorical variables, e.g. trailer type, braking system type and other ‘type’ variables, would be better presented/easier to interpret if shown in their categories/frequencies – rather than medians. Even if it was easier just to show the top three most frequently reported (rather than all types, if there are many.)

Response: In response to the comments of two of the four reviewers, the tables have been edited in include medians and IQR. The details of the categories for each research question have been provided by inclusion of the original questionairres in the supplementary materials provided with  the manuscript.

Line 217. The figure 7/16 would seem to refer to just one of the horse behaviours (scrambling?) rather than both scrambling and panicking – which are currently the subject of the sentence. This sentence should be revised either to reflect both behaviours (eg x/28) or just refer to scrambling.

Response: Than you. This has now been clarfied in the text.

Line 239. Are you suggesting that there is less representation from SA and Vic because of a (relative) lack of internet connection? I’m not sure I agree, if this is what you are saying. Perhaps consider reviewing this sentence or make it clearer what point you mean here.

Response: Following the comments of Reviewer 1, lines 234- 243 have now been deleted

Line 330. Typo ‘to the on the dashboard’

Response: Following the comments of Reviewer 1, lines 326- 332 have now been deleted

Reviewer 2 Report

There are errors in the simple summary that should be corrected. In line 16 remove the numeral '1'.
In line 21 remove horses between transportation and injuries.
In line 28 warrant should be warrants
in line 25 it would be clearer to say that "There was a trend for participants who reported less than 8 hours sleep in the night before transport to report injury."

Author Response

Thank you for your assistance in improving this manuscript. 

Please see specify responses below.

There are errors in the simple summary that should be corrected. In line 16 remove the numeral '1'.

Response: Adjusted in text – thank you. The simple summary has been corrected and adjusted in response to the comments of reviewers.

In line 21 remove horses between transportation and injuries.

Response: Adjusted in text – thank you.

In line 28 warrant should be warrants

Response: Adjusted in text – thank you.

in line 25 it would be clearer to say that "There was a trend for participants who reported less than 8 hours sleep in the night before transport to report injury."

Response: The statement as written is correct. The finding was associated with the driver, and not the injury incident. The related question was designed to solict information for further factors based study, rather that to prove the behaviour as directly causative of the injury to the horse. This has now been further clarified in the text.   

Reviewer 3 Report

This paper addresses an interesting topic.  However, in its present format is has a number of major limitations.  The title does not reflect the real content of the paper and the methods used leave some level of uncertainty about the time frame and the period at risk for the outcome variable.  The confounding of these variables means that the data needs to be reanalysed so these are accounted for and clear definition of the period at risk provided.

Specific comments

Line 16 typo delete l before movement

Line 42 this is the first flag about the questions regarding analysis and interpretation of the data (r2 = 0.2) this mean 80% of the variance is due to other factors, and this is univariable, so some caution should be used when presenting this regression value.

Lines 58-60 this is taken out of context. It may be relevant with the commercial transport of horses to slaughter, but not for the commercial transport of horses not for slaughter –please rewrite to avoid bias and misrepresentation.

Line 67-68 please rewrite or rephrase this is a little misleading – particularly with respect to trucks as both private and commercial transporters have the same level of compliance standards

Line 93-94 could you provide some clarification around the time at risk period for the injury data reported.  Was this within the last year, or lifetime.  The lack of clarity around this provides some uncertainty about the validity of the analysis.  There is a real need to provide this – or correct the injury data for some measure of time at risk.  Given the sampling frame time at risk (hours in truck and number of horses in truck and lifetime travel) will differ considerable between participants at equitana vs mount lofty pony club eventing – this is a major flaw and needs to be addressed within your outcome variable (horse injury).

Statistical analysis

Was there any correction for multiple comparisons?

Line 149-150 I was a little bemused to see truck make in the analysis.  Surely gross vehicle mass or tare weight, number of axles and number of horses carried are of more importance and provide better differentiation than brand of truck as this tells us nothing of capacity or risk.

Table 2

Why given the skewed nature of driver age and experience are these presented as mean and SD (surely this failed the normality test) when others are median and range – also please use regular convention and present median and interquartile range – this is more useful for the reader.

Line 157-158 – please identify clearly the time at risk for when you asked questions regarding injury – if lifetime then major bias due to recall and this needs to be addressed and discussed

Line 165 please rewrite the current structure is misleading …There were no significant …..

Line 177-179 no difference between brands so why mention

Line 181 This shows the flaws in the analysis if time at risk is not accounted for.  If you carry more horses and travel greater distances you have a greater relative time at risk for a horse in jury --- this needs to be addressed in defining your outcome variable. 

Figure 1 why present simple a univarable correlation which is so low – correct for time at risk and then include this in the multivariable model to account for the confounders  – then this may be worth presenting .

Line 193 again the authors need to be careful of not placing too much emphasis on the screening univarable outcomes.

Lines 200-204 – as above caution should be used with univariable analysis – was this significant  ?

 Line 218-220 why not just say no breed effect

Table 5

This appears to imply the more you transport a horses the greater the time at risk and chance of an injury.  Please analyse data accounting for time at risk

Discussion

As a rule the discussion is verbose and lacking in focus

Lines 231-238 sorry but I lost the logic and flow of this first section of discussion

Line 235-238 a little confused here why talk about demographics in Australia and use a UK reference?

Lines 291 – 295 an illogical argument

Line 306 – is this the first time RTA is introduced?  – if so please present in full initially

Line 325 – 332 please rewrite - conjecture and a bit of a red herring

Line 33-336 see earlier comments on the issue of not placing too much emphasis on univariable analysis – these all fell out of the multivariate model – so not significant for your study

Line 243-253 discussion is losing focus – delete or rewrite

Lines 353-359 live animal towing licence seems bureaucratic and heavy handed- if not logistically impractical to implement and manage.  Please justify – is this used in other countries – or are other education models in place ?

Author Response

Thank you for your assistance in improving the manuscript

Reviewer 1

This paper addresses an interesting topic.  However, in its present format is has a number of major limitations.  The title does not reflect the real content of the paper and the methods used leave some level of uncertainty about the time frame and the period at risk for the outcome variable.  The confounding of these variables means that the data needs to be reanalysed so these are accounted for and clear definition of the period at risk provided.

Response: The authors have addressed the reviewer’s specific comments below. Recognising that the study does not reflect a true estimate of risk in the statistical sense, the word “Risk” has been omitted from the title.

Specific comments

Line 16 typo delete l before movement

Response: Adjusted in text – thank you.

Line 42 this is the first flag about the questions regarding analysis and interpretation of the data (r2 = 0.2) this mean 80% of the variance is due to other factors, and this is univariable, so some caution should be used when presenting this regression value.

Response: The authors agree with the reviwer. To that end the discussion of this finding is not extensive, and refers to it’s modest significance only.

Lines 58-60 this is taken out of context. It may be relevant with the commercial transport of horses to slaughter, but not for the commercial transport of horses not for slaughter –please rewrite to avoid bias and misrepresentation.

Response: The authors have considered the reviewer’s remark carefully and reexamined the cited refrences, and the reference lists used in these articles. Both of the references cited encompass discussion of the welfare of horses transported for slaughter, as well as for general haulage and ron-commercial pruposes. The word”commercial” has bee removed from line  58 in consideration of the reviwer’s concern.   

Line 67-68 please rewrite or rephrase this is a little misleading – particularly with respect to trucks as both private and commercial transporters have the same level of compliance standards

Response: The authors agree that the intent of this phrase is imprecise, and have rewritten this statement and made reference to the relevant legislation in Australia.

Line 93-94 could you provide some clarification around the time at risk period for the injury data reported.  Was this within the last year, or lifetime.  The lack of clarity around this provides some uncertainty about the validity of the analysis.  There is a real need to provide this – or correct the injury data for some measure of time at risk.  Given the sampling frame time at risk (hours in truck and number of horses in truck and lifetime travel) will differ considerable between participants at equitana vs mount lofty pony club eventing – this is a major flaw and needs to be addressed within your outcome variable (horse injury).

Response: The authors appreciate the reviewer’s comments, and that this lack of clarity may affect the readers’ interpretation of the results. Although this is clear in the survey question reported attcahed in the supplementary materials, this has now been clarified in the methods and materials section. Information summarising the time that has elapsed between the reported injury and the time of the survey has been appended to the results section. The limitations of recall bias and its effects on the validity of results in a study of this nature are now discussed and referenced towarsd the end of the paper. Given the relatively small “n” for injuries, the authors appreciate limitations of any statistical analysis of these data. We have cautioned the reader later in the discussion, and have now attempted to strengthen that caution. Nevertheless the observations made during this small scale study may assist other researchers (and ourselves) in formulating more precise questions and determining avenues of investigation for what is clearly a complex subject. The authors respectfully submit that the stated aims for the study articulate the rather conservative aspirations of the group in their first foray into this field. It was not a stated aim of the current study to estimate true period risk. Such a complex study was beyond the resources and time available to the research team.

The authors agree with the reviewer that some of the more complex issues that he/she has raised do require validation, but are certainly beyond the scope of this current data set. However the (hard) lessons learnt that were facilitated by this small scale study will better inform our future work, and hopefully that of others. To that end we have already commenced some experimental research, driven by these data, that is more focused in teasing out the relative significance of some of the factors that were found to be “associated” with injury.

Statistical analysis

Was there any correction for multiple comparisons?

Multiple incidents were not documented in the study.

Line 149-150 I was a little bemused to see truck make in the analysis.  Surely gross vehicle mass or tare weight, number of axles and number of horses carried are of more importance and provide better differentiation than brand of truck as this tells us nothing of capacity or risk.

Response: The authors agree. Initially it was the intent of the data gathering team to collect such data. As alluded to in the materials and methods section (lines 98-102), this became impractical. Basically our research team was not large enough to cope with the volume of respondents and vehicles at these events. Future studies that involve direct interviews may need a larger team if they are to encompass the collection of a largef dataset and more vehicle. We are still curious about factors such as floor area, tare, engine capacity and other features and whether or not they affect the risk.

Table 2

Why given the skewed nature of driver age and experience are these presented as mean and SD (surely this failed the normality test) when others are median and range – also please use regular convention and present median and interquartile range – this is more useful for the reader.

Response: The authors agree with this suggestion. The data was re-plotted and Shapiro Wilk W tests performed (not sure how this was missed previously) . These data are indeed not normally distributed. We have reported the median and interquartile range. However because of possible concerns about welfare issues were identified by one of the other reviewers, we have also included the full range in a separate column.

Line 157-158 – please identify clearly the time at risk for when you asked questions regarding injury – if lifetime then major bias due to recall and this needs to be addressed and discussed

Response: As indicated above, the authors agree. This is clarified in the methods and materials section, and results, and discussed as a study limitation.

Line 165 please rewrite the current structure is misleading …There were no significant …..

Response: Adjusted in text.

Line 177-179 no difference between brands so why mention

Response: A good point-thanks. These data have now been omitted.

Line 181 This shows the flaws in the analysis if time at risk is not accounted for.  If you carry more horses and travel greater distances you have a greater relative time at risk for a horse in jury --- this needs to be addressed in defining your outcome variable.

Response: With respect, the authors found this comment somewhat difficult to understand. The key outcome variable for comparative analyses was injury (or noninjury). The authors agree that it is likely that the number of horses carried does not solely explain the risk of injury in isolation. Surface area (density), ventilation, age and gender mix, duration of journey and many other factors are likely to contribute to the risk. Although some of these factors have been evaluated by other researchers, the authors are not aware of any statistical modelling around all of these variables for horse transportation. Such modelling was certainly not possible with the current limited sample size for injury.

Figure 1 why present simple a univarable correlation which is so low – correct for time at risk and then include this in the multivariable model to account for the confounders  – then this may be worth presenting .

Response: Based on the recall bias, and the limitations with the dataset is unlikely that accurate or meaningful correction for time at risk is possible. The relationship may provide soft support for further investigation around the role of age, maintenance and design. It was not the author’s intention to overemphasise this finding, and agree that the figure is best removed.

Line 193 again the authors need to be careful of not placing too much emphasis on the screening univarable outcomes.

Response: Respectfully these lines are simple descriptive reporting of the data. No particular emphasis is given or intended.

Lines 200-204 – as above caution should be used with univariable analysis – was this significant  ?

Response: Fatigue has been demonstrated as one of the most important risk factors associated with motor vehicle injury. It is not the intent of the authors to overstate this finding. However we do consider it worth noting, as it may encourage others to more robustly investigate this as a possible risk factor.

Line 218-220 why not just say no breed effect

Response: A good point-thanks. These data have now been omitted. The names of the breeds compared have retained as they are one of the factors of interest to the horse owning public.

Table 5

This appears to imply the more you transport a horses the greater the time at risk and chance of an injury.  Please analyse data accounting for time at risk

Response: The data collected would result in fewer than 35 injured horses being available if the time at risk were calculated and included as suggested. This falls to 24 injuries when of those responses that were provided within the range of 10 to 15 or 15 to 20 years are omitted (as described in the survey question; supplementary materials). The authors accept that the data set is insufficient for substantial modelling. Within the constraints of the dataset the authors have explored modelling, understanding the likelihood of more than two variables found to be significant would be low (based on basic epidemiologic principles on the ratio of observations relative to the number significant variables). Recognising these limitations, the authors will consider advice as to whether the multivariable modelling should be included or excluded from the paper.

Discussion

As a rule the discussion is verbose and lacking in focus

Lines 231-238 sorry but I lost the logic and flow of this first section of discussion

Response: Omitted.

Line 235-238 a little confused here why talk about demographics in Australia and use a UK reference?

Response: Sorry – a citation error.

Lines 291 – 295 an illogical argument

Response: The study cited investigated smaller non-articulated vehicles, and is mentioned for comparison and contrasting purposes. The statements in this paragraph have been simplified and reworded.

Line 306 – is this the first time RTA is introduced?  – if so please present in full initially

Response: No. First mentioned and defined in line 264 of the original version of the manuscript.

Line 325 – 332 please rewrite - conjecture and a bit of a red herring

Response: This statement is a reflection on the authors’ current research direction, and agree with the reviwer that it does not constitute a disussion of the resuts of the current paper (i.e. a red herring”). This has been ommitted.

Line 33-336 see earlier comments on the issue of not placing too much emphasis on univariable analysis – these all fell out of the multivariate model – so not significant for your study

Response: As suggested by the reviewer, this section has been carefully reworded to reflect the survey findings, without overemphasis.

Line 243-253 discussion is losing focus – delete or rewrite

Response: The authors assume that the reviewer intended to refer to lines 343-353. The authors agree that this paragraph does not add substantial value to the manuscript. It has been deleted.

Lines 353-359 live animal towing licence seems bureaucratic and heavy handed- if not logistically impractical to implement and manage.  Please justify – is this used in other countries – or are other education models in place ?

Response: In agreement with other workers in the field, the authors are oncerned about the current lack of training of inexperienced persons in the transporation of horses in Australia. The data collected on drivers in this study reinforced  those concerns. The comments have been toned down, and reference to current UK towing licences, and possible commercially oriented training  that may possibly be adapted  to this purpose referenced.

Reviewer 4 Report

The article aims to provide information about the risk of injuries in horses transported under non-commercial conditions.

Since for this a survey was conducted I have some doubts in relation to how a major and minor injury was defined when the survey was applied, It would help the reader if more information is provided on this, since drivers may overlook injuries that a veterinarian would classify as minor. 

In the survey it is asked how long have drivers been driving. Is there a legal limit in Australia?, since the survey goes up to 12h it seems to be a long time in relation to what is allowed in other countries. Do drivers have to stop in between and allow horses to rest?. Is food or water provided during long journeys? are horses restrained within the trucks?.

In the simple summary please eliminate the "l" before movement in line 16.

Although it is explained why measures of the trucks and trailers were not obtained, having an average space availability for horses during transport could provide relevant information in relation to injuries, the same with position of horses within the vehicle.

In the results section: Line 165, the beginning of the sentence can be confusing when starting with A significant association was not found, maybe clearer to state No significant association was found.

Line 171-173, "corresponding values for trailer..." are this results still in relation to timing? (loading, unloading, etc) a little confusing.

Line 180. When talking about "horse carrying capacity", does it mean that at higher capacity the horses had more space? and this could affect balance?, or are they restrained and even at higher capacity the space allowance would be the same? or is there actually another horse in the vehicle?

Line 196. In relation to "answering the phone", is this allowed in Australia while driving? or does this imply using a handless device? this should be specified.

In relation to drivers experience, it would have been interesting to have information about any courses on horse handling and not only about driving experience. Is there any certification system for live-animal transport during commercial transport?.

The conclusions are clear and the limitations of the study are also recognized within the manuscript, such as the limited sample size. I believe after minor clarifications the manuscript is acceptable for publication.

Author Response

Thank you for your assistance with our manuscript.

The article aims to provide information about the risk of injuries in horses transported under non-commercial conditions.

Since for this a survey was conducted I have some doubts in relation to how a major and minor injury was defined when the survey was applied, It would help the reader if more information is provided on this, since drivers may overlook injuries that a veterinarian would classify as minor.

Response: The concerns of the reviewer are noted.  The survey was an owner based one and the owner oriented definitions of each are provided in the survey tools used. In the case of owners and veterinarians, the classification as to whether injuries are minor moderate or marked is likely to be highly subjective, as there are no objective criterion. Indeed recent evidence suggests that owners frequently assess and treat injuries without veterinary consultation.1.2  In order to respond to the reviewers questions would be necessary to conduct a study that further examined owners perceptions and there ability to assess injuries. This was beyond the scope of the current study , but may be of interest in future investigations.

1.     Theoret CL, Bolwell CF, Riley CB. A cross-sectional survey on wounds in horses in New Zealand. New Zealand Veterinary Journal 2016;64:90-94.

2.     Sole A, Bolwell CF, Dart A, Riley CB, Theoret CL. Descriptive survey of wounds in horses presented to Australian veterinarians Australian Equine Veterinarian 2015;34 (4): 68-74.

In the survey it is asked how long have drivers been driving. Is there a legal limit in Australia?, since the survey goes up to 12h it seems to be a long time in relation to what is allowed in other countries. Do drivers have to stop in between and allow horses to rest?. Is food or water provided during long journeys? are horses restrained within the trucks?.

Response: The authors’ knowledge there are no legal limits to the length of time a driver may operated vehicle noncommercially (there are strict limits for commercial drivers). The  Australian Standards and Guidelines for the Welfare of Animals – Livestock, Transport version 1.1 26 September 2011 provides for the land transport of horses. The degree will to which the public involved in the non-commercial transport of horses in Australia are aware of these guidelines has not, to the author’s knowledge, been ascertained. It is also equally unclear as to what level these have been enforced with the non-commercial movement of horses. Horses are committed to be transported up to 36 hours. The conditions under which this is permissible are contained within the guidelines. The means by which horses are restrained in the trucks or float was not included as a question in the survey. However it may be a useful consideration in future surveys. (http://www.australiananimalwelfare.com.au/app/webroot/files/upload/files/ID100%20SG%20for%20Horses.pdf).

In the simple summary please eliminate the "l" before movement in line 16.

Response: Adjusted in text – thank you.

Although it is explained why measures of the trucks and trailers were not obtained, having an average space availability for horses during transport could provide relevant information in relation to injuries, the same with position of horses within the vehicle.

Response: The authors agree. The authors have commenced other studies to investigate some of these factors in a controlled experimental setting.

In the results section: Line 165, the beginning of the sentence can be confusing when starting with A significant association was not found, maybe clearer to state No significant association was found.

Response: Text altered as suggested-I thank you

Line 171-173, "corresponding values for trailer..." are this results still in relation to timing? (loading, unloading, etc) a little confusing.

Response: The description of these failures has been clarified in text

Line 180. When talking about "horse carrying capacity", does it mean that at higher capacity the horses had more space? and this could affect balance?, or are they restrained and even at higher capacity the space allowance would be the same? or is there actually another horse in the vehicle?

Response: The text has been expanded to make this clearer, and more closely reflect the intent of the survey question. As the trasnportation vehicles were not examined by the researchers, this is what the owner believed to be the capaicty of their own vehicle. The ratio of floor space unfortunately could not be reported in the current study, as measurement parameters of the more than 46 plus trailer models and the trucks were not available.

Line 196. In relation to "answering the phone", is this allowed in Australia while driving? or does this imply using a handless device? this should be specified.

Response: This practice is legal in all states but must be operated entirely hands-free or placed in a cradle to place a call, use a GPS or play music. The survey did not investigate whether respondents and the phone in hands-free or “in-hand” mode.

In relation to drivers experience, it would have been interesting to have information about any courses on horse handling and not only about driving experience. Is there any certification system for live-animal transport during commercial transport?.

Response: The authors did seek information about any training that the non-commercial drivers had received. None indicated they had completed and such these courses. Although commercial transport was not subject to the current study there are Australian Welfare Standards and Guidelines that require training and documentation of that training by employers involved in the commercial transport of livestock, but no formal certification. Lines 259-261 of the original manuscript make statements that reflect some of the interests expressed above by the reviewer.

The conclusions are clear and the limitations of the study are also recognized within the manuscript, such as the limited sample size. I believe after minor clarifications the manuscript is acceptable for publication.

Response: Thank you

Round 2

Reviewer 1 Report

The authors have adequately addressed my earlier concerns and the manuscript now accurately reflects the data.

Reviewer 3 Report

General comments to authors

I am afraid a number of the important questions over the structure and presentation of the data have not been adequately responded to.  The responses indicate a lack of understanding of the limitations of the data and the way the data has been interpreted.  My recommendation would be the authors rewrite / present the data as a simple descriptive study.  The small dataset and the very extended period over which they are asking for information presents significant recall bias and in places there is over interpretation of the data.  The authors have failed to identify that given the sampling time – up to 20 years – that in this study there was a low real frequency / risk of injury or incident.

Specific comments to authors

Original Line 42 – this has still not been addressed – this demonstrates over interpretation of the data.  This unfortunately is a theme that reappears throughout the paper.

Original line 58-60

The response to this query is poor.  There are significant differences in the structure of the vehicles, the care of stock and mixing of stock with transport to slaughter that means there is no great similarity with personal transport of horses in horse floats / trailers.

Line 89 spp – experienced

Lines 101-103  This explanation demonstrates a major flaw with the data collected and how it has been interpreted.  The lack of time frame constraints means some data collected is from a very long time period (up to 20 years ago!).  This has a major impact on how the authors describe risk – as the analysis later on clearly shows the more time travelling and distance travelled the greater the odds were of reporting having had an incident. 

Table 1 is a bit misleading as the number of surveyed with data does not reflect the data presented in the table.  Please include a column of valid surveys from these events.

Line 156 truck brands are still listed – why ? what is the purpose.

Table 2.  This table demonstrates a lack of consistent data collected with the surveys.  Please discuss this and impact in the discussion.

Why are we lacking data on how many years ago the incident the sampling time is a major concern with the validity and robustness of the data reported.  Given the very large sampling timeframe and the low number of incidents reported then the risk appears very low – this does not seem to be mentioned in the discussion at all with the authors focusing on injury and avoidance of it, when it would appear it is of such a low frequency it may be hard to reduce frequency.

Also the authors have only reported the denominator as the trailer incidents  (44 vs 55 events in the lines above)– were truck injuries / incidents now not included in the data.  These inconsistencies are a concern

Line 189 Reporting brands is of no interest to non WA readers, just report 3 main brands and multiple single representations.

Line 193 – sorry but p=0.09 is a big step with this data to imply a trend. Please delete

Line 196 was this corrected age of trailer at incident – or age of trailer at sampling.  Also please see earlier comments regarding this regression should this be  r2 here – this is not a modest relationship – standard nomenclature for this is weak relationship.

Lines 208-219 There needs to be considerable discussion about the limitations in this data due to recall bias.  I think most would agree that the time frame for recall greatly limits, if not invalidates, a lot of the implied precision in some of the data reported here.  

Binary logistic regression

I had severe doubts over the validity of this analysis and interpretation.  After checking with a number of colleagues they also independently identified this is flawed and should not have been conducted with this data given the way it was collected and time at risk.